# Intracellular Membrane Contact Sites in Skeletal Muscle Cells

**DOI:** 10.3390/membranes15010029

**Published:** 2025-01-14

**Authors:** Matteo Serano, Stefano Perni, Enrico Pierantozzi, Annunziatina Laurino, Vincenzo Sorrentino, Daniela Rossi

**Affiliations:** 1Department of Molecular and Developmental Medicine, University of Siena, 53100 Siena, Italy; matteo.serano@unisi.it (M.S.); stefano.perni@unisi.it (S.P.); enrico.pierantozzi@unisi.it (E.P.); annunziatina.laurino@unisi.it (A.L.); vincenzo.sorrentino@unisi.it (V.S.); 2Program of Molecular Diagnosis of Rare Genetic Diseases, Azienda Ospedaliera Universitaria Senese, 53100 Siena, Italy

**Keywords:** endoplasmic reticulum, sarcoplasmic reticulum, calcium signaling, muscle contraction

## Abstract

Intracellular organelles are common to eukaryotic cells and provide physical support for the assembly of specialized compartments. In skeletal muscle fibers, the largest intracellular organelle is the sarcoplasmic reticulum, a specialized form of the endoplasmic reticulum primarily devoted to Ca^2^^+^ storage and release for muscle contraction. Occupying about 10% of the total cell volume, the sarcoplasmic reticulum forms multiple membrane contact sites, some of which are unique to skeletal muscle. These contact sites primarily involve the plasma membrane; among these, specialized membrane contact sites between the transverse tubules and the terminal cisternae of the sarcoplasmic reticulum form triads. Triads are skeletal muscle-specific contact sites where Ca^2^^+^ channels and regulatory proteins assemble to form the so-called calcium release complex. Additionally, the sarcoplasmic reticulum contacts mitochondria to enable a more precise regulation of Ca^2^^+^ homeostasis and energy metabolism. The sarcoplasmic reticulum and the plasma membrane also undergo dynamic remodeling to allow Ca^2^^+^ entry from the extracellular space and replenish the stores. This process involves the formation of dynamic membrane contact sites called Ca^2^^+^ Entry Units. This review explores the key processes in biogenesis and assembly of intracellular membrane contact sites as well as the membrane remodeling that occurs in response to muscle fatigue.

## 1. Introduction

The endomembrane system of eukaryotic cells consists of a network of organelles that provide compartmentalization for distinct cellular functions. These organelles include the nuclear envelope, the endoplasmic reticulum (ER), the Golgi apparatus, mitochondria, and lysosomes. Initially regarded as separate entities, it has become evident over the past decades that these organelles are dynamically interconnected through well-defined membrane contact sites (MCSs). At these sites, opposing organelles are tethered at a distance of 10–50 nm without fusing. The formation of MCSs requires the involvement of tethering proteins, shaping proteins, and cytoskeletal elements, which assemble into tethering complexes that serve both structural and functional roles (reviewed by Voeltz, et al. [1]). Following their discovery, the functional roles of MCSs have significantly expanded. Functions previously thought to rely on vesicular transport or molecular diffusion can be now at least in part attributed to MCSs. These processes include the transfer of lipids, Ca^2^^+^, amino acids, and metal ions; the exchange of signaling molecules that regulate mitochondrial and endosomal fission; the biogenesis of autophagosomes and lipid droplets; and the regulation of lipid synthesis (reviewed by Scorrano, et al. [2]).

For the purposes of this review, it is noteworthy that the first MCSs were observed in striated muscle cells in the 1950s between the ER—referred to as the sarcoplasmic reticulum (SR)—and transverse tubules (TTs), which are specialized regions of the plasma membrane (PM) [3]. Additional MCSs were later identified in muscle cells, connecting the SR with other organelles, such as mitochondria, lipid droplets, and lysosomes. More recently, MCSs that do not involve the SR have also been discovered.

This review focuses on the organization and composition of the SR, the assembly of intracellular MCSs between the SR and other organelles, and the membrane remodeling that occurs in response to muscle fatigue.

## 2. The SR: Morphology, Composition, and Function

In eukaryotic cells, the ER consists of an elaborate network of membranes composed of a lipid bilayer of phospholipids and sterols, to which peripheral and transmembrane proteins are associated. Morphologically, the ER can be divided into distinct structural domains: in the central part of the cell, ER tubules and sheets expand from the nuclear envelope and distribute throughout the entire cell volume to reach the cell periphery [4,5,6]. These domains are highly dynamic and constantly undergoing remodeling: ER tubules undergo fusion and branching resulting in the generation of new junctions, while ER sheets with variable sizes can assemble in a single layer or with a stacked conformation, connected by twisted membranes with helical edges [7]. These structural variations are supported by lipid composition, intrinsic membrane curvature, association with specific ER shaping proteins, and binding with microtubules facilitating the formation of specialized functional domains that support the different ER functions: ER sheets usually host ribosomes on their cytosolic surface and are thus referred to as rough ER, where protein translation, translocation, post-translational modifications, and folding can occur; on the contrary, the high curvature of ER tubules is incompatible with ribosome attachment and they thus form the so-called smooth ER that is primarily associated with lipid synthesis, Ca^2^^+^ storage, and signaling [8].

The first description of the ER in skeletal muscle cells by Porter and Palade in 1957 revealed the existence of a particular vacuolar system that was originally referred to as the sarcoplasmic reticulum (SR) [3]. The elegant description of the SR depicted a continuous reticular configuration organized as a sleeve-like structure around myofibrils that regularly segmented in phase with the striations of the associated sarcomeres, a disposition that prompted the supposition that the SR was functionally important in muscle contraction [3,9]. Indeed, the SR can be clearly distinguished in two highly specialized subdomains composed of tubules and cisternae or sheets: the tubular domain forms the longitudinal SR (l-SR), arranged around the M- and Z-lines of each sarcomere, and is mainly devoted to Ca^2^^+^ uptake thanks to the presence of sarcoplasmic/endoplasmic reticulum Ca^2^^+^ ATPase (SERCA) pumps and their regulatory proteins like phospholamban and sarcolipin; sheet-like structures form the junctional SR (j-SR), consisting of flat cisternae localized, in mammals, at the border between the A- and I-bands of each sarcomere and facing regular invagination of the sarcolemma, the TTs. The structure formed by two terminal cisternae and one TT is called a triad (Figure 1). The j-SR is responsible for Ca^2^^+^ release in response to nerve stimulation and thus allocates the components of the Ca^2^^+^ release complex, including the ryanodine receptor type 1 (RYR1), triadin, junctin, and calsequestrin [10,11,12]. Mutations in these proteins are linked to various forms of human myopathies [13,14,15,16].

Unlike the ER in other eukaryotic cells, the l-SR and j-SR represent stable domains that do not undergo remodeling and maintain a stable association with the sarcomere. Indeed, the SR juxtaposition around myofibrils is guaranteed by the interaction between the small muscle-specific isoform of ankyrin 1 (sAnk1.5) localized on the l-SR membrane and obscurin, a giant protein localized in the sarcomere [17,18,19,20]. Accordingly, mice knockouts for either of these two proteins show a significant reduction in the volume of the l-SR around the myofibrils [21,22,23,24].

The SR occupies about 10% of the total cell volume. Its lipid composition is similar to that of the ER in other eukaryotic cells, with phosphatidylcholine (approximately 60%) and phosphatidylethanolamine (about 15%) as major components, along with minor contributions from phosphatidylserine, sphingolipids, phosphatidylinositol, and cholesterol. Phosphatidylcholine and phosphatidylethanolamine have asymmetric distributions, with the former found mostly in the inner leaflet and the latter in the outer leaflet [25]. Differences in lipid composition between fast-twitch and slow-twitch muscles or during development and aging have been described [26,27,28,29]. Van Winkle and collaborators also suggested that differences in the ratio between saturated and unsaturated phospholipids may exist when comparing the longitudinal and the junctional domains of the SR [30].

Additional components of the SR can be found at the neuromuscular junctions, where different structures can be observed. In the rat diaphragm, at the base of the membranous folds of the sarcolemma, small and flat cisternae are found; on the contrary, in the subsarcolemmal region rich in mitochondria, the SR is mainly composed of larger cisternae. Finally, between the postsynaptic folds, the SR is formed by thin tubes running close to the PM [31].

Although the term SR was originally used to generally indicate the ER of muscle cells, the SR was later described as a specialization of the smooth ER unique to muscle cells, functionally dedicated to the regulation of Ca^2^^+^ storage and release. This has sometimes caused confusion, further complicated by incomplete understanding of where the additional functions of the SR, not related to Ca^2^^+^ handling necessary for muscle contraction, are carried out in muscle cells. Nevertheless, today, it is recognized that the terms ER and SR refer to a single membrane system, where distinct functional subdomains can be identified, hence also the use of the term ER/SR.

Immunofluorescence studies on the localization of ER markers in the SR showed that calnexin and the translocon components sec61α, sec61β, and TRAPα colocalize with SERCA in the l-SR around the Z disk/I-band, suggesting that microdomains of resident ER proteins responsible for protein translocation, glycosylation, and folding exist within this domain [32,33,34]. More recently, the presence of ribosomes and protein synthesis were confirmed to be localized around the Z disk/I-band, with a minor fraction of mRNA also present around the myonuclei [35]. Localization of ER to Golgi exit sites has also been proposed to reside in the perinuclear and the subsarcolemmal regions. Interestingly, ER markers show different solubilization properties that may be explained by the lipid composition of surrounding membranes, thus suggesting that lipid microdomains may drive the segregation of these proteins within the l-SR [32,33,36]. The existence of lipid microdomains in the SR, and in general in the ER of eukaryotic cells, however, remains an area of investigation. The low cholesterol content in the ER, estimated at approximately 3–5% of total ER lipids [37], appears to argue against the formation of lipid rafts similar to those found in the PM. Nevertheless, some studies suggest that nanodomains resembling functional lipid rafts can exist at ER exit sites assisting the transport of glycosylphosphatidylinositol (GPI)-anchored proteins to the Golgi [38]. In addition, recent research on S-Palmitoylation, a post-translational modification that promotes binding of proteins to lipid-raft domains in the PM, has revealed that palmitoylation of a cysteine 678 in the j-SR protein junctophilin-2 (JPH2) helps stabilize the protein’s anchoring within the SR membrane [39].

The presence of a quality control system of protein synthesis has also been observed in the ER/SR, where accumulation of misfolded proteins can activate the Unfolded Protein Response (UPR), which attenuates the translation machinery, preventing further accumulation of misfolded proteins, and/or induce ER-Associated Degradation (ERAD) [40,41]. Activation of the UPR can also produce deleterious effects. Chronic SR stress contributes to skeletal muscle loss [42,43], and alterations of the UPR or ERAD were observed in some forms of myopathies. In SELENON-related myopathies, mutations or ablation of SELENON results in activation of a maladaptive UPR mediated by CHOP and the disulphide oxidase endoplasmic oxidoreductin 1 (ERO1), which result in muscle damage and wasting [44,45]. On the contrary, adaptive proteostasis mechanisms, like those regulated by the protein complex composed of SEL1L and the E3 ubiquitin ligase HRD1, were found to be critical for managing not only misfolded proteins in skeletal muscle but also muscle development and growth [46].

## 3. SR and Plasma Membrane Contact Sites: The Triad

### 3.1. The Triad: Structure and Function

The skeletal muscle triad is the first and most thoroughly studied membrane contact site. This specialized structure is critical for muscle contraction by enabling the cross-communication between proteins on the SR and those residing on the TTs [47].

In striated muscle cells, the PM—often referred to as the sarcolemma—undergoes significant remodeling and specialization, a process equally vital to the optimization of skeletal muscle contractility as the specialization of the SR [48].

Structurally, the triad is composed of two enlarged SR vesicles, known as terminal cisternae, positioned on either side of a central TT (Figure 1 and Figure 2). This arrangement represents one of the most striking examples of a structure–function relationship in subcellular structures, to the extent that the functional role of the triad in muscle contraction could be hypothesized, with reasonable accuracy, from the early ultrastructural observations [3]. Functionally, the triad is also referred to as a “calcium release unit” (CRU) or “couplon”, underscoring its critical role in the process that allows the translation of the action potential, propagated along the sarcolemma, into Ca^2^^+^ release from the SR, ultimately triggering muscle contraction. This process is known as excitation–contraction coupling (ECC). The triad’s critical function is rooted in its unique architecture, which allows a precise juxtaposition between the SR, serving as the main intracellular Ca^2^^+^ reservoir, and the TT, which facilitates the rapid transmission of action potentials deep into the muscle fiber.

### 3.2. The TT: Composition and Biogenesis

The T-tubular system’s ability to form isolated vacuoles under osmotic shock has enabled the purification of inside-out TT vesicles and the analysis of their membrane composition. These membranes contain higher levels of cholesterol, sphingomyelin, and acidic phospholipids, such as phosphatidylserine and phosphatidylinositol, compared to SR membranes. Despite a reduction in phosphatidylcholine compared to SR vesicles, this phospholipid remains the most abundant in TT membranes [49,50]. Unsurprisingly, the lipid composition of TTs closely resembles that of the sarcolemma, with which they are in direct continuity. A notable difference, however, is the lower protein-to-lipid ratio observed in TTs compared to the sarcolemma [49,51,52,53].

According to their functional role in skeletal muscle, TTs are enriched in the muscle-specific Ca_V_1.1 voltage-gated Ca^2^^+^ channels, which represent the main subunit of the dihydropyridine receptor (DHPR), responsible for sensing variation in the muscle fiber’s membrane depolarization. Approximately 80% of membrane DHPRs are localized within TTs, particularly at SR-TT junctions [54] Other T-tubular proteins include caveolin 3 (CAV3), cavin 4, and dynamin 2, which are involved in early TT formation [55,56], as well as bridging integrator-1 (BIN-1) and myotubularin 1 (MTM1). BIN-1 plays a pivotal role in TT biogenesis and elongation and, through its ability to induce membrane curvature, may serve as a diffusion barrier restricting the exchange of proteins between the sarcolemma and TTs [57,58]. MTM1 is a phosphoinositide 3-phosphatase that specifically hydrolyzes PI(3)P and PI(3,5)P [59], two phospholipids critical for recruiting FYVE domain-containing endosomal proteins. These lipids are important for lipid exchange between the ER/SR and TTs, and for the proposed role of MTM1 in the stabilization of mature TTs [59,60,61,62,63].

Although early morphological studies provided valuable insights into the timing and structural organization of TT maturation, the molecular mechanisms underlying the initial stages of TT biogenesis remain an active area of investigation and have only recently begun to be elucidated. Initial hypotheses proposed three possible mechanisms to explain the rapid expansion of the T-tubular network: inward elongation of the PM [64], addition of membranes through vesicle fusion [65,66], and the formation of an internal network that ultimately fuses with the PM [67]. In 2020, Hall and colleagues, using live imaging of zebrafish embryos, 3D electron microscopy, and computational modeling, provided evidence supporting Ishikawa’s hypothesis of inward elongation [55]. Their model suggests that TTs originate from regions of the sarcolemma containing clathrin-independent endocytosis markers, such as CD44, EH domain-containing 1 (EHD1), and dynamin 2. These elongating tubules are stabilized through interactions with the developing SR system and with sarcomeric proteins like titin. The elongation process is supported by the endocytic pathway and requires a substantial increase in membrane synthesis [55]. In a follow-up study [68], the same research group demonstrated that BIN-1 colocalizes with muscle-specific proteins CAV3 and cavin 4 at early TT nucleation sites. BIN-1′s N-BAR domain forms a concave dimer that binds to membrane lipids, thus inducing the membrane curvature necessary for TT inward elongation. Additionally, BIN-1′s SH3 domain interacts with cavin 4, which, in turn, recruits CAV3-positive caveolae. As TTs elongate, cavin 4 and CAV3 are recycled back to the sarcolemma in a process dependent on both BIN-1 and cavin 4.

In a more recent study, Lemerle and colleagues expanded on the work of Lo, Hall, and collaborators by exploring the earliest stages of TT formation in unroofed murine and human myotubes [56]. They observed BIN-1-decorated ring-like structures on the inner leaflet of the sarcolemma, which recruited CAV3-positive caveolae, forming a “pearled ring”. The caveolae-enriched regions of the ring serve as nucleation sites for early TT development. Further elongation is supported by the skeletal muscle-specific BIN-1 splice variant in which exon 11 is retained [69]. Interestingly, this elongation occurs without the continued involvement of cavin 4 and CAV3-positive caveolae, which are removed in temporal coincidence with the formation of SR/TT junctions, suggesting that caveolin removal is crucial for junction formation. The proposed mechanism of TT formation and elongation can, thus, be summarized in the following steps: (1) formation of the subsarcolemmal ring structures by BIN-1; (2) recruitment of cavin 4/CAV3-positive caveolae to the ring; (3) nucleation of the TT and inward elongation supported by the endocytic pathway and mediated by muscle-specific BIN-1, cavin 4, and CAV3; (4) removal of caveolae from the TT surface, facilitated by cavin 4; (5) association of the TTs to the developing SR and to structural sarcomeric proteins.

### 3.3. SR and TT Maturation, Striving for the Triad

The triadic junction is so essential for muscle development and function that the remodeling of both TT and SR is centered around its formation.

The triad is the culmination of a complex process of remodeling, elongation, and association of the developing SR and sarcolemma which starts in embryonic life (Figure 3) (reviewed in [70,71,72]). The development of the SR has been studied during the embryonic stage in chick and rodent skeletal muscles. At days E10-E11 of chick development, new SR membranes start to develop from the rough ER and connect with one another to form a network around the myofibril [73]. At E12, the SR is well developed around the Z-line and I-band of the sarcomeres [70]. Similarly, in mammal embryonic muscles at E12-E14 the SR begins to form “peripheral junctions” in which cortical SR cisternae align with regions of the PM at the surface of the fiber. In newborn rats, the l-SR still incompletely surrounds the myofibrils, with irregularly oriented tubules running along consecutive sarcomeres, and the j-SR is still composed of sparse flat cisternae connected with SR tubules. The maturation of the SR reaches completion only after the full development and final arrangement of the TT system, which is accomplished during the following 7-10 days after birth [74,75].

The development of the TT starts later than that of the SR (Figure 3). Early T-tubular structures appear around day E14, coinciding with the emergence of subsarcolemmal myofibrils. By E16, nearly all muscle fibers display a well-developed TT network, predominantly oriented along the longitudinal axis of the fiber [71,75,76]. At this stage, the TT network intricacy is enriched by short transversally oriented interconnections, loops, and bends arising from occasional associations with the developing SR system. As mentioned above, the gradual rearrangement of the TTs from longitudinal to transverse starts at E17 and reaches completion only postnatally at 7-10 days of age [75,76], but the association between TTs and the SR commences much earlier. A transition from peripheral couplings into internal TT-SR junctions begins as early as E15 and reaches completion at E18 [75]. Since at this stage, the TTs are still in their longitudinal orientation, the internal junctions are mostly oriented longitudinally or, occasionally, force the junctional region of the TT to assume a local transverse orientation. Interestingly, the preferential localization of these immature internal junctions at the A-I-band interface occurs prior to the final maturation of the T-tubule, suggesting that the precise positioning of the triad in this location is likely driven by the sarcoplasmic reticulum (SR) and its docking to sarcomeric structural proteins, rather than by the TT [75].

### 3.4. Maturation of the Triad

A fully mature triad can be identified by three distinct ultrastructural features. First, in mammals, triads are located at the interface between the A-band and I-band in the internal regions of muscle fibers. Second, they are composed of two terminal cisternae of the SR aligned along the longitudinal plane of the fiber, flanking a central TT oriented perpendicular to the fiber axis. Third, the triad displays regularly arranged electron-dense structures, known as “feet,” bridging the gap between the terminal cisternae and the TT. Each of these feet represents RYR1, a large tetrameric Ca^2^^+^ channel embedded in the j-SR membrane and responsible for releasing Ca^2^^+^ from the SR lumen into the cytoplasm.

The maturation of the triad occurs in two distinct phases (Figure 3). The first phase coincides with the development of the SR. Between E14 and E16, a gradual change in the organization and localization of CRUs is observed. Initially, CRUs are primarily located at the cell periphery, forming peripheral couplings or shallow junctions with short, wide invaginations of the PM. Over time, these peripheral couplings are replaced by internal junctions, which initially consist of a single SR cisterna flanking a TT (a structure known as a dyad), and later acquire a second SR cisterna to form the complete tripartite triad complex [75]. As seen for the SR, there is a significant acceleration in triad maturation between E16 and E18, as peripheral couplings disappear entirely, and internal junctions become the predominant structural feature. During this critical period, CRUs shift from a random distribution across sarcomeric regions to a more localized positioning at the A-I-band interface. This transition is also marked by an increasing presence of RYR1 feet within the junctional gap. At E14 and E15, these feet are sparse or absent, by E16, about half of the SR junctional surface is covered with RYR1 feet, and this proportion increases to approximately 80% by E18 [75]. The second phase of maturation mirrors the final development of the TT system, beginning at birth and continuing through the first two weeks of life. By the end of this period, triads present with the TT oriented perpendicular to the fiber axis and longitudinally disposed SR cisternae.

### 3.5. Triadic Proteins

Being the subcellular microdomain responsible for converting electrical signals into intracellular Ca^2^^+^ release, the triad contains ion channels that are directly involved in voltage sensing and Ca^2^^+^ release, along with a variety of other triadic proteins that regulate Ca^2^^+^ homeostasis and provide structural support to stabilize this MCS [77]. The key players in ECC are the DHPRs, specifically their α1S subunit, the Ca_V_1.1 channel, in the TT, and the RYR1 in the j-SR membrane. The α1S subunit is a voltage-gated Ca^2^^+^ channel that senses depolarization of the sarcolemma, while RYR1 is a high-conductance Ca^2^^+^ channel that releases large amounts of Ca^2^^+^ from the SR into the cytoplasm, initiating muscle contraction. These two channels are physically coupled [78], allowing the conformational change induced by membrane depolarization in DHPR to be mechanically transmitted to RYR1, leading to its opening. This “conformational coupling” is unique to skeletal muscle and facilitates the rapid conversion of the action potential along the muscle fiber membrane into intracellular Ca^2^^+^ release, allowing for high-frequency stimulation required for skeletal muscle contraction during intense activity. Although there is strong evidence supporting this conformational coupling [79,80], the exact nature of the interaction between DHPR and RYR1 remains unclear. However, it is known that this process functionally depends on at least two additional proteins: the skeletal muscle-specific β1a subunit of DHPR [81,82] and the adaptor protein “SH3 and cysteine rich domain 3” (Stac3) [83,84]. The absence of any of these proteins results in ECC ablation and perinatal death, while loss or gain of function mutations lead to a myopathic phenotype [81,83,85,86,87,88,89,90]. From a structural standpoint, a third protein, junctophilin (JPH), is also essential for ECC [91]. JPHs were identified in the early 2000s when they were first cloned from skeletal muscle (JPH1 and JPH2), cardiac muscle (JPH2), and neurons (JPH3 and JPH4) [92,93]. Each member of this family anchors to the SR/ER membrane through their C-terminal domains and binds electrostatically to the plasma membrane via their N-terminal regions. In skeletal muscle, both JPH1 and JPH2 play key roles in the formation and stabilization of triads, although their expression timelines differ significantly. JPH2 expression occurs as early as embryonic day 14 (E14) and peaks by E17, whereas JPH1 expression is only detectable postnatally [94]. This suggests that JPH2 primarily facilitates the formation of CRUs during early muscle development, while JPH1 contributes to the later stages of triad maturation. Indeed, mice lacking JPH1 are able to breathe and move, but die shortly after birth due to suckling defects [94]. Their muscles show reduced triad junctions, abnormal SR, lower contractile force, and elevated reliance on extracellular Ca^2^^+^. These findings indicate that while JPH2 alone can support ECC, JPH1 is crucial for the full maturation of triads and may play a more prominent role in the development of specific muscle groups, such as the jaw, pharyngoesophageal, and diaphragm muscles. Investigating JPH1′s ability to support ECC independently in mouse models is challenging because JPH2, the sole isoform expressed in the heart, is essential for cardiac ECC, and JPH2 knockout mice die in utero [93]. However, recent studies have demonstrated that JPH1 can sustain voltage-induced Ca^2^^+^ release in tsA201 cells even in the absence of JPH2 [95]. Beyond their structural role, junctophilins also influence the molecular composition of the junctional domain by binding and recruiting specific channels to both the ER/SR and PM sides of the junction [95,96,97].

## 4. SR and Plasma Membrane Contact Sites: The Calcium Entry Unit (CEU)

As previously mentioned, the ER also functions as an intracellular Ca^2+^ reservoir playing a pivotal role in many cellular processes due to the critical importance of this ion in cell homeostasis and signaling. In all cells, and especially in muscle cells, the ER undergoes extensive remodeling and specialization to meet the adequate Ca^2+^ requirements of the cell. In resting conditions, cytoplasmic Ca^2^^+^ levels are maintained at very low concentrations, with the primary sources of Ca^2^^+^ being the extracellular space and intracellular compartments or organelles that sequester and store cytoplasmic Ca^2^^+^. The Ca^2^^+^ concentration in the ER is kept at steady levels to support the various Ca^2^^+^-dependent cellular processes. A key system that evolved to ensure the replenishment of the intracellular Ca^2^^+^ stores is represented by Store-Operated Ca^2^^+^ Entry (SOCE) [98]. This ubiquitous process was first identified in salivary gland cells [98]. Decades of intensive research have revealed that SOCE is primarily governed by the interaction of two critical proteins: stromal interaction molecule 1 (STIM1), a single-pass transmembrane protein located on the ER membrane, and Ca^2+^ release-activated calcium modulator 1 (ORAI1), a Ca^2^^+^ release-activated channel (CRAC) on the PM. In resting cells, STIM1 adopts a folded conformation and is diffusely distributed along the ER membrane, where it interacts with the microtubule-associated protein EB1 [99,100]. When the luminal Ca^2^^+^ content of ER decreases, Ca^2^^+^ detaches from low-affinity binding sites in the luminal domain of STIM1. This triggers a conformational change in STIM1, which causes the protein to oligomerize and detach from EB1, finally resulting in its translocation to the PM. The association of STIM1 with the PM is likely due to the structural rearrangement of Ca^2^^+^-unbound STIM1, which exposes the C-terminal polybasic domain, thus allowing binding to phospholipids like PI(2)P and the interaction with the Ca^2^^+^ channel ORAI1 [101,102,103] within STIM1′s CRAC activation domain, facilitating the gating of ORAI1, enabling Ca^2^^+^ influx from the extracellular space. The entire sequence—from ER Ca^2^^+^ depletion to Ca^2^^+^ entry activation—occurs within tens of seconds, with STIM1’s translocation to the PM being the time-limiting step [104,105]. This process also involves extensive remodeling of the ER, forming elongated cortical ER cisternae that are visible in electron micrographs [106,107]. The junctions are dynamic and highly transient; neither STIM1 nor ORAI1 permanently reside at these sites. Instead, intracellular Ca^2+^ store depletion triggers a dynamic reorganization, bringing STIM1 and ORAI1 together at junctional regions to assemble the SOCE complex [108]. To date, the precise mechanisms underlying the coordinated translocation of these two proteins across two distinct membrane compartments remain unclear.

Not surprisingly, given the pivotal role of Ca^2^^+^ signaling in regulation of muscle contraction, SOCE mechanisms in skeletal muscle have adapted to efficiently support Ca^2^^+^ entry, ensuring an adequate amount of Ca^2^^+^ within the SR lumen. Accordingly, two STIM isoforms, STIM1 and STIM2, are expressed in human skeletal muscle [102,109,110,111,112].

STIM1 is the primary regulator of SOCE in skeletal muscle, while STIM2, with lower Ca^2^^+^ affinity and slower activation kinetics, plays a less characterized role [110,113]. The muscle-specific STIM1 splice variant, STIM1L, includes 106 additional amino acids that tether it to cortical actin. This facilitates STIM1L’s stable association to the PM, enabling fast SOCE activation and supporting the rapid Ca^2^^+^ cycling needed to maintain muscle function [109,114]. Both STIM1 and STIM1L are able to interact with ORAI1 and transient receptor potential cation channels (TRPCs), providing further levels of modulation of Ca^2^^+^ entry. An additional splice variant, STIM2.1, can heterodimerize with STIM1 to negatively regulate SOCE, adding complexity to the system [115]. In skeletal muscle, STIM1 localizes across all the sarcomeres. Indeed, STIM1 is present at the triad, as indicated by its colocalization with RYR1, but it is also observed in the l-SR [116]. The functional role of l-SR-located STIM1 is unclear, but it may represent a reserve pool mobilized to enhance SOCE, as discussed in the following section, or perform alternative functions such as activating SERCA1 [117]. Three isoforms of ORAI protein (ORAI1, ORAI2, and ORAI3) are expressed in skeletal muscle tissue [111,118,119,120,121,122,123].

### SR and TT Remodeling: Assembling the Calcium Entry Units

In adult skeletal muscle fibers, TTs and j-SR at triads form stable MCSs. More recent evidence has shown the remodeling capability of SR membranes in correspondence to the I-band. In resting muscles, the SR at the I-band presents as a complex network that appears as vesicles and convoluted tubular structures in longitudinal sections. However, after exercise, these membranes may reorganize into stacks of flat, multiple parallel layers of cisternae composed of two or more elements. This membranous remodeling also involves the TT, which extends toward the stacks of SR [124]. It was proposed that these structures might be involved in Ca^2^^+^ homeostasis and specifically act as SOCE sites for replenishing local depletion of SR Ca^2^^+^ during intense muscle activity [125]. This hypothesis is supported by a significant increase in colocalization between STIM1 and ORAI1 observed following treadmill exercise. Indeed, prior to exercise, STIM1 is spread throughout the sarcomere with minimal overlap with ORAI1, which presents an almost exclusive triadic localization. After exercise, a significant portion of ORAI1 relocalizes to the I-band, where STIM1 is found, enhancing their colocalization [124]. Consequently, muscles from exercised animals exhibit greater resistance to fatigue during repetitive stimulation compared to controls. This increased resistance is eliminated when extracellular Ca^2^^+^ entry is blocked, either by using a Ca^2^^+^-free bath solution or SOCE inhibitors, indicating the critical role of Ca^2^^+^ entry in this effect [124,125,126,127,128]. Additionally, it was demonstrated that Ca^2^^+^ entry facilitated by SR and TT remodeling at the I-band augments Ca^2^^+^ transients, resting cytoplasmic Ca^2^^+^ levels, and contractile force during repetitive stimulation [125,129,130]. These results strongly suggest that the SR and TT remodeling observed following intense exercise represents Calcium Entry Units (CEUs) where SOCE occurs [131] (Figure 2). Accordingly, both SOCE and assembly of CEUs were absent in two distinct Orai1-knockout models [129,130,132].

Recent research aiming to unravel the mechanism leading to CEU formation demonstrated that functional CEUs can be detected ex vivo following repeated stimulation of EDL muscles. This suggests that CEU assembly does not require innervation or blood supply but is intrinsic to the muscle itself [133]. Nevertheless, the molecular mechanisms underlying such TT rearrangement remain unknown, nor is it clear whether this represents a universal mechanism of SOCE during normal muscle function.

The observed membrane remodeling associated with intense exercise currently seems to be peculiar to skeletal muscle fibers, as in other cell types the establishment of ER/PM contact sites dedicated to SOCE primarily appears to rely on ER remodeling promoted by STIM1 and/or by tethering proteins such as junctate or E-synaptotagmin-1, with limited changes in PM morphology [134].

Interestingly, the presence of CEUs, along with SR remodeling, has been observed in mice lacking calsequestrin 1, calsequestrin 2, or both [113,135,136]. One might speculate that the absence of calsequestrin, the primary Ca^2^^+^ binding protein in the SR lumen, could lead to a reduced SR Ca^2^^+^ load, causing the muscle fiber to rely more heavily on extracellular Ca^2^^+^ entry via SOCE.

## 5. SR and Mitochondria Contact Sites

Mitochondria–endoplasmic reticulum (ER) contact sites (MERCs) play critical roles in numerous cellular processes and are supported by protein tethers including the inositol (1,4,5)- trisphosphate receptor (IP(3)P-R) and mitofusin 2 (MFN2), in the ER, and the glucose-regulated protein 75 and anion channel (GRP75), the voltage-dependent anion channel 1 (VDAC1), or either mitofusin 1 (MFN1) or MFN2 in the mitochondrial membrane [137,138,139,140,141,142,143] (Figure 2). MERCs support essential functions such as the transfer of Ca^2^^+^ from the ER to the mitochondrial matrix to regulate ATP production and lipid exchange between the two organelles [140,144,145,146,147]. Delivery from the ER to mitochondria can also control induction of apoptosis under conditions of cellular stress [148,149].

In skeletal muscle, mitochondria account for approximately 20% of the cell volume and are mostly organized as an interconnected network [150,151]. Mitochondria are divided into two different populations: the subsarcolemmal mitochondria, round in shape and localized beneath the sarcolemma, close to the nuclei; and the intermyofibrillar mitochondria, which appear more elongated and localized close to the triads [152]. Subsarcolemmal mitochondria are suggested to have a greater capacity to drive oxidative phosphorylation and distribute ATP to the mitochondrial network, whereas interfibrillar mitochondria primarily support localized energy demand for muscle contraction [153].

Extensive contact sites are established between the SR and mitochondria [154]. In mouse skeletal muscle, mitochondrial distribution and MERC frequency change during development: early in development, mitochondria are longitudinally arranged and move toward a triad-adjacent location in mature muscle fibers. The number of tethers increases during the first four postnatal months [155,156]. Recent 3D imaging studies showed that, at birth, MERCs are present, although the mitochondrial networks are disorganized and poorly associated with the contractile apparatus. Following postnatal muscle maturation, MERCs in non-triadic SR regions decrease and mitochondria located in these regions become more closely associated with lipid droplets [157].

Tethers between the j-SR and the outer mitochondrial membranes have been observed with an average length of 10 nm; tethering proteins have been mainly studied in cardiac muscle, where a 50 KDa isoform of MFN2 has been proposed to be involved in MERC formation [158]. In contrast, the role of full-length MFN2 in MERC formation is still debated since knockout of MFN2 apparently does not affect MERC assembly [159]. Evidence of interaction between RyRs and VDAC has also been provided [160,161]. Interestingly, the Ca^2^^+^ conductance of the mitochondrial Ca^2^^+^ uniporter (MCU) (a low-affinity Ca^2^^+^ transport protein of the inner mitochondrial membrane) was found to be much higher in skeletal muscle than in other cell types, suggesting a local Ca^2^^+^ transfer from the SR to the mitochondrial matrix via alignment of VDAC and MCU [162,163,164]. MERCs in skeletal muscle are functionally linked to mitochondrial Ca^2^^+^ uptake in response to increased sarcoplasmic Ca^2^^+^ concentration during muscle contraction and to regulation of ATP production [154,165,166,167]. Elevated mitochondrial Ca^2^^+^ activates Ca^2^^+^-sensitive matrix dehydrogenases (including pyruvate dehydrogenase phosphatase, isocitrate dehydrogenase, and oxoglutarate dehydrogenase) as well as ATP synthase, enhancing ATP production during sustained muscle activity [168]. Interestingly, changes in MERCs have also been associated with muscle metabolic requirements: an increase in MERCs and mitochondrial fusion are associated with ATP production from fatty acid oxidation, while a decrease in MERCs and increase in mitochondrial fission are associated with elevated glycolytic activity [169].

Additional functions of MERCs in skeletal muscles are progressively emerging. Recent findings show that the UPR-induced protein kinase R (PKR)-like endoplasmic reticulum kinase (PERK) and endoribonuclease/protein kinase IRE1-like protein (IRE1) are located at MERCs [170]. Adaptive UPR signaling via MERCs can increase mitochondrial Ca^2^^+^ import, metabolism, and dynamics, while maladaptive UPR can result in activation of apoptotic pathways [171]. MERCs also support phospholipid exchange between the ER and mitochondria, a process mediated by members of the ER membrane protein complex (EMC) [172]. Muscle-specific EMC1 knockdown was shown to cause defective locomotion and shortened life span, associated with muscular deformities, altered SR network, and impaired mitochondrial oxidative phosphorylation in Drosophila [173].

## 6. SR and Lysosome Contact Sites

Lysosomes are acidic organelles responsible for degradation of damaged or excess intracellular components to maintain cellular homeostasis; they also degrade extracellular components following endocytosis. Beyond their recycling roles, lysosomes also regulate intracellular lipid homeostasis by interacting with other organelles such as the ER, lipid droplets, and mitochondria, enabling direct lipid transfer at MCSs [1].

In cardiac and skeletal muscle, lysosomes also support additional functions, including Ca^2^^+^ buffering [174]. Accordingly, impairment of lysosomal function has been associated with disorders of the cardiac rhythm or heart failure [175,176,177]. In atrial and ventricular myocytes, lysosomes are located near the SR and the mitochondria supporting the assembly of Ca^2^^+^ signaling nanodomains between these organelles [178]. Release of Ca^2^^+^ from lysosomes occurs through type 2 two-pore channels (TPC2) after activation by nicotinic acid adenine dinucleotide (NAADP) [175]. Interestingly, CD38, the enzyme that catalyzes the synthesis of NAADP, has been shown to localize to an SR domain close to lysosomes [179]. In cardiomyocytes, release of Ca^2^^+^ from lysosomes indirectly increases the amplitude of SR Ca^2^^+^ transients during action potential; Ca^2^^+^ release from lysosomes is mirrored by an increased uptake of Ca^2^^+^ into the SR that was proposed to be mediated by CaMKII phosphorylation of phospholamban; this would result in greater SR Ca^2^^+^ content and thus increased amplitude of Ca^2^^+^ transients. An alternative explanation proposes that RYR2, reported to be enriched at lysosome–SR contact sites, can be directly activated by a Ca^2^^+^-induced Ca^2^^+^ release mechanism [175,180,181]. Supporting this, knockout of TPC1 and TPC2 reduces SR Ca^2^^+^ release, mainly due to decreased RYR2 expression [182]. However, the precise protein composition of lysosome–SR contact sites remains debated, since some studies reported that RYR2 is not present at these contact sites in cardiac muscle [178], while in smooth muscle, NAADP-induced Ca^2^^+^ release from lysosomes was proposed to induce local Ca^2+^ transients due to activation of clusters of RYR3 [183,184].

## 7. SR, Lipid Droplets, and Mitochondria

Lipid droplets store lipids for energy production, membrane biogenesis, and synthesis of signaling molecules [185]. They consist of neutral lipids, primarily triacylglycerol and sterol esters, surrounded by a phospholipid monolayer, where specific proteins like lipases are localized [186]. Contact sites between lipid droplets and ER are essential for lipid droplet formation and budding and for the removal of damaged proteins from the ER [186].

Lipid droplets are also extensively interconnected with the so-called peridroplet mitochondria, facilitating fatty acid supply to mitochondria for energy requirements [187]. In skeletal muscles, both subsarcolemmal and intermyofibrillar lipid droplets are present [188]. About 70% of lipid droplets in these regions are estimated to be in contact with mitochondria via perilipin-5 (PLIN5) tethering [186,189].

Interestingly, in brown adipose tissue, peridroplet mitochondria exhibit unique metabolic properties including higher ATP synthesis capacity, reduced fusion–fission dynamics, lower fatty acid oxidation capacity, and increased tricarboxylic acid cycle activity. This suggests that they represent a subpopulation of mitochondria with specific energetic functions [187]. Whether this subpopulation is also present in skeletal muscle remains unclear.

## 8. SR and the Nuclear Envelope

The nuclear envelope is composed of an outer nuclear membrane facing the cytoplasm and an inner nuclear membrane that faces the nucleoplasm. The outer nuclear envelope is continuous with the ER membranes that are also partially connected with the inner nuclear membrane at the edge of the nuclear pore complexes. The perinuclear space between the outer and the inner nuclear membrane is also continuous with the ER lumen [190]. Interestingly, at the points of connection between the nuclear envelope and the ER, the curvature of the nuclear membrane is particularly evident, forming constrictions where exchange of lipids and proteins between the two organelles is facilitated [191]. The nuclear envelope also forms the so-called nucleoplasmic reticulum, a network of tubules that extends inside the nucleoplasm that can be divided in type I (tubules originating from the inner nuclear envelope) and type II (tubules originating from both the inner and the outer nuclear envelope) membranes [192]. Continuity between the perinuclear space and the lumen of the ER enables an extensive exchange of lipids and proteins, but also signaling molecules, including Ca^2^^+^. Indeed, the presence of IP(3)P-Rs and RYRs in the nuclear envelope has been proposed in different cell types including cardiomyocytes [193,194,195,196], as well as that of STIM1, ORAI1, and SERCA [196,197]. The D84G mutation in STIM1, which results in constitutively active SOCE, is associated with expansion of the perinuclear space, altered nuclear morphology and mechanical properties, and changes in the perinuclear Ca^2^^+^ content, suggesting a novel role for STIM1 as a sensor for mechanical stress and nuclear Ca^2^^+^ store content [198]. These findings suggest that the nuclear envelope is equipped with Ca^2+^ handling proteins and that it may thus actively participate in intracellular Ca^2^^+^ signaling, acting either autonomously or in coordination with the SR. Indeed, studies in cardiomyocytes showed that the SR lumen is continuous with the perinuclear space, forming a large shared Ca^2^^+^ storage compartment [196], and that cytosolic Ca^2+^ transients can be mirrored by nuclear Ca^2+^ transients [199]. The regulation of Ca^2^^+^ signaling in the nuclear membrane in skeletal muscle, however, is still poorly understood.

The SR cisternae are also continuous with the nuclear membrane at the neuromuscular junction, where IP(3)P-Rs have been found to mediate the release of Ca^2^^+^ from intracellular stores upon activation of acetylcholine receptors according to a Ca^2^^+^-induced Ca^2^^+^-release mechanism [200,201,202] (Figure 2).

## 9. Conclusions

The ER/SR is a single-copy organelle characterized by a continuous membrane structure, which extends for a long distance throughout the entire length of a muscle fiber. While its role as a Ca^2^^+^ storage organelle in the context of muscle contraction has been extensively studied, recent research highlights that the SR is also engaged in multiple MCSs with other intracellular organelles, creating a complex interplay and functional interdependence among them. Indeed, Ca^2^^+^ fluxes not only regulate muscle contraction but also modulate ER stress by interacting with chaperones such as BiP/GRP78, calnexin, and calreticulin. Additionally, they influence muscle bioenergetics by regulating mitochondrial ATP production and affect gene expression by modulating expression or activity of transcription factors. Within this intricate interplay, individual organelles remain actively interconnected, being engaged in extensive interactions while maintaining their structural and functional integrity.

The challenge of future years will be to understand how the SR can segment its membrane into multiple functional microdomains and further elucidate the structure of MCSs in skeletal muscle cells. The use of advanced imaging techniques, such as volume or three-dimensional (3D) correlative light and electron microscopy (volume-CLEM), could reveal unexpected and novel three-dimensional structures of the endomembrane system in muscle cells, offering surprising new insights into its organization and function.

## Figures and Tables

**Figure 1 membranes-15-00029-f001:**
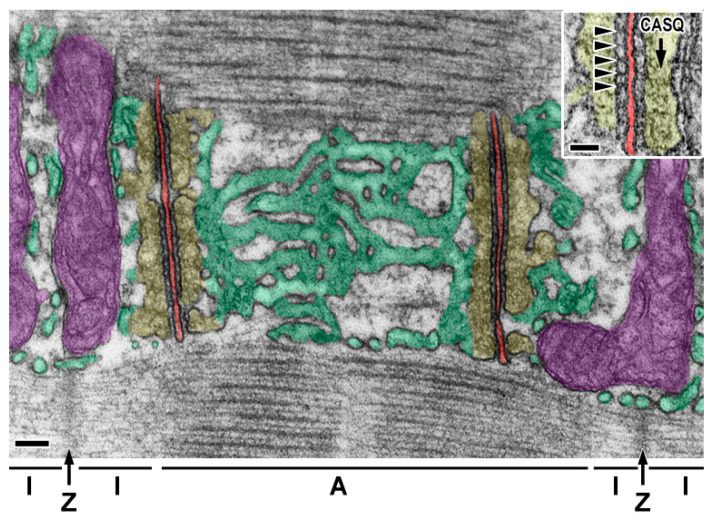
Micrograph of a mouse flexor digitorum brevis muscle showing a portion of a myofibril’s surface, which reveals the SR in face view. The l-SR, pseudo-colored in green, forms a tubular network along the sarcomere’s A-band. Toward the center of the A-band, the l-SR tubules merge into flat, fenestrated cisternae. The terminal cisternae of the junctional SR (j-SR), pseudo-colored in yellow, are visible at the boundary between the A-band and I-band. These cisternae contain an electron-dense, mesh-like substance identified as polymerized CASQ (indicated by the arrow in the inset). Two j-SR cisternae flank a central T-tubule, pseudo-colored in red, which runs perpendicular to the myofibril’s longitudinal axis, forming the triad. Mitochondria are highlighted in purple. The lines and letters below the main figure indicate the sarcomeric I-band (I), Z-line (Z), and A-band (A). The inset provides a high-magnification view of the triad, showing the RYR1 feet as electron-dense structures (marked by arrowheads) bridging the junctional gap between the j-SR and the T-tubule. Scale bars: 100 nm and 50 nm (inset).

**Figure 2 membranes-15-00029-f002:**
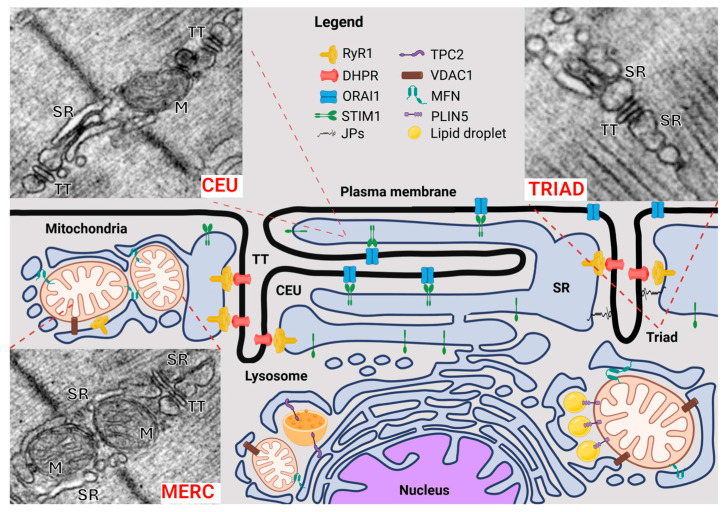
Schematic representation of membrane contact sites (MCUs) between subcellular membranous organelles in skeletal muscle cells. The illustration highlights key interaction sites, including sarcoplasmic reticulum–T-tubule contact sites such as the triad and Calcium Entry Unit (CEU), mitochondria–sarco/endoplasmic reticulum contacts (MERCs), lysosome–sarco/endoplasmic reticulum contacts, nucleus–sarco/endoplasmic reticulum contacts, and lipid droplet–sarco/endoplasmic reticulum contacts. Key proteins with structural and/or functional roles in these MCUs are also depicted. Abbreviations: SR = sarcoplasmic reticulum; TT = T-tubule; M = mitochondria. Created with BioRender.com.

**Figure 3 membranes-15-00029-f003:**
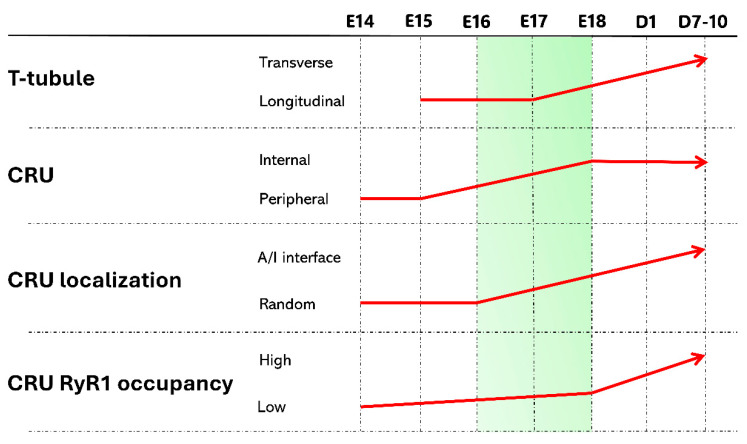
Time dependency of structural remodeling driving the maturation of the SR, TT, and CRUs in skeletal muscle. The arrows illustrate the timeline of membrane reorganization occurring from embryonic day 14 (E14) to postnatal days 7–10. The slope of each arrow’s oblique segment approximatively reflects the relative rate of the corresponding process. The highlighted E16-E18 period represents a critical period for the acceleration of SR development and the beginning of full T-tubule maturation. Adapted from [75].

## Data Availability

No new data were created.

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
