# Peer review of "Intracellular Membrane Contact Sites in Skeletal Muscle Cells"

_membranes, 2025, doi:10.3390/membranes15010029_

Round 1

Reviewer 1 Report

Comments and Suggestions for Authors

This research review focuses on the sarcoplasmic reticulum (SR) in skeletal muscle cells, describing its structure, composition, and diverse membrane contact sites with other organelles. The SR's role in calcium signaling and muscle contraction is explained. The review also examines dynamic MCSs, such as calcium entry units formed during muscle fatigue, and the SR's interactions with mitochondria and lysosomes, highlighting their roles in energy metabolism and cellular homeostasis. Finally, the review discusses the developmental processes involved in the maturation of the SR and its MCSs. The review is very extensive, and well written. It a very interesting read, I learned a lot and strongly recommend publishing it. I only have a few minor comments that the authors might consider as I believe it would further increase the quality of their review.

The paragraph in lines 80-100 describes the arrangement of the sarcoplasmic reticulum, and its organization in L-SR, j-SR relative to A and I bands. This description is quite complex and difficult to follow for non-experts; an additional figure would help to understand this.

The organization of the SR/ER membrane into functional membrane domains is only very briefly described in lines 145-148. Can the authors perhaps explain this in a bit more detail. What exactly is the proposed mechanism, and what is the evidence for this? As the membrane contains only very little cholesterol, are these domains also thought to be driven by cholesterol?

The same goes for the description of the ultrastructural features of the mature triad described in lines 291-298. Can the authors provide a figure explaining the arrangement of the A-band, I-band, terminal cisternae, and electron dense “feet”?

I appreciate the clear diagram in Figure 1. However, the insets showing the electron microscopy images are less clear. Can the authors annotate these, or provide additional diagrams, to indicate what the various structures in the MERC, CEU, TRIAD are?

I spotted several minor typo’s:

-Line 67: “remodeling” should be “remodel”

-Lines 386, 390, 400, 483, 524, and 551, the plus symbol of Ca2+ not in superscript

-Line 440: space missing “ORAI1observed”

Author Response

Reviewer 1 Comments for the Author:

This research review focuses on the sarcoplasmic reticulum (SR) in skeletal muscle cells, describing its structure, composition, and diverse membrane contact sites with other organelles. The SR's role in calcium signaling and muscle contraction is explained. The review also examines dynamic MCSs, such as calcium entry units formed during muscle fatigue, and the SR's interactions with mitochondria and lysosomes, highlighting their roles in energy metabolism and cellular homeostasis. Finally, the review discusses the developmental processes involved in the maturation of the SR and its MCSs. The review is very extensive, and well written. It a very interesting read, I learned a lot and strongly recommend publishing it. I only have a few minor comments that the authors might consider as I believe it would further increase the quality of their review.

The paragraph in lines 80-100 describes the arrangement of the sarcoplasmic reticulum, and its organization in L-SR, j-SR relative to A and I bands. This description is quite complex and difficult to follow for non-experts; an additional figure would help to understand this.

We thank the reviewer for the suggestion and for appreciating our manuscript. To better depict the organization of the sarcoplasmic reticulum we added a new figure (now figure 1) illustrating the arrangement of the longitudinal and junctional domains of the sarcoplasmic reticulum with respect to the sarcomere. We hope that this new figure may help in better understand the organization of the sarcoplasmic reticulum in skeletal muscle.

The organization of the SR/ER membrane into functional membrane domains is only very briefly described in lines 145-148. Can the authors perhaps explain this in a bit more detail. What exactly is the proposed mechanism, and what is the evidence for this? As the membrane contains only very little cholesterol, are these domains also thought to be driven by cholesterol?

The organization of functional domains within the SR/ER membrane remains an active area of investigation. In response to the reviewer’s suggestion, additional points of discussion have been added to the text to address this topic, being aware that many questions in this field remain unresolved (lines 137-147).

The same goes for the description of the ultrastructural features of the mature triad described in lines 291-298. Can the authors provide a figure explaining the arrangement of the A-band, I-band, terminal cisternae, and electron dense “feet”?

New figure 1 has been designed to meet the reviewer suggestion to improve the description of the ultrastructural features of the mature triad.

I appreciate the clear diagram in Figure 1. However, the insets showing the electron microscopy images are less clear. Can the authors annotate these, or provide additional diagrams, to indicate what the various structures in the MERC, CEU, TRIAD are?

Figure 1 (now figure 2 in the revised manuscript) has been modified according to the reviewer suggestion: the various structures have been now more clearly labeled, and the figure legend has been changed.

I spotted several minor typo’s:

-Line 67: “remodeling” should be “remodel”: The sentence has been corrected to “constantly undergoing remodeling”

-Lines 386, 390, 400, 483, 524, and 551, the plus symbol of Ca2+ not in superscript: the typo has been corrected

-Line 440: space missing “ORAI1observed” : the space has been added

Reviewer 2 Report

Comments and Suggestions for Authors

Matteo Serano et al. systematically explored the structure, biogenesis, function, and dynamic remodeling of intracellular membrane contact sites in skeletal muscle cells, with a particular focus on key structures such as triads, Calcium Entry Units (CEUs), and Mitochondria-ER Contact Sites (MERCs). The paper is comprehensive, with detailed citations and significant value for research in this field. While the paper is well-structured and informative, there is still room for improvement in language expression, formatting, and scientific details. The following are specific suggestions for revision:

  1. There are significant issues with the formatting of calcium ions throughout the paper. In some instances, there is no space after the ion (e.g., in the abstract), and in others, the "+" sign is not superscripted. This is a fundamental error and requires careful correction.
  2. The paper lacks sufficient summary figures. It is recommended to add 1-2 additional summary figures or tables.
  3. On page 3, line 42: The phrase “can be now be at least in part attributed” contains an extra “be” that needs to be removed.
  4. On page 8, line 248: While the spelling of “curvature” is correct in this instance, the term is frequently misspelled as “curvatrure” elsewhere in the text.
  5. On page 11, line 395: Regarding the description of the SOCE mechanism, “occurs within tens of seconds,” it is recommended to provide original experimental data to support this time range.
  6. On page 13, line 523: The description of mitochondrial calcium regulation conflicts with the explanation of nuclear membrane calcium signaling on page 15, line 611. These functional relationships should be clarified further.

Author Response

Reviewer 2

Comments and Suggestions for Authors

Matteo Serano et al. systematically explored the structure, biogenesis, function, and dynamic remodeling of intracellular membrane contact sites in skeletal muscle cells, with a particular focus on key structures such as triads, Calcium Entry Units (CEUs), and Mitochondria-ER Contact Sites (MERCs). The paper is comprehensive, with detailed citations and significant value for research in this field. While the paper is well-structured and informative, there is still room for improvement in language expression, formatting, and scientific details. The following are specific suggestions for revision:

  1. There are significant issues with the formatting of calcium ions throughout the paper. In some instances, there is no space after the ion (e.g., in the abstract), and in others, the "+" sign is not superscripted. This is a fundamental error and requires careful correction.

We apologize for the formatting errors. We have thoroughly reviewed the text and corrected all the typos

  1. The paper lacks sufficient summary figures. It is recommended to add 1-2 additional summary figures or tables.

A new figure (now figure 1) has been added to illustrate the arrangement of the longitudinal and junctional domains of the sarcoplasmic reticulum with respect to the sarcomere. We hope this addition enhances the reader's understanding of the sarcoplasmic reticulum's organization in skeletal muscle and improves the overall clarity of the manuscript.

  1. On page 3, line 42: The phrase “can be now be at least in part attributed” contains an extra “be” that needs to be removed.

The extra “be” has been deleted.

  1. On page 8, line 248: While the spelling of “curvature” is correct in this instance, the term is frequently misspelled as “curvatrure” elsewhere in the text.

The text has been reviewed.

  1. On page 11, line 395: Regarding the description of the SOCE mechanism, “occurs within tens of seconds,” it is recommended to provide original experimental data to support this time range.

References (104 and 105) to original papers describing the time range of SOCE have been added.

  1. On page 13, line 523: The description of mitochondrial calcium regulation conflicts with the explanation of nuclear membrane calcium signaling on page 15, line 611. These functional relationships should be clarified further.

We apologize for any confusion in these two points of discussion. We have revised the text to clarify that mitochondrial Ca²⁺ uptake primarily results from localized increases in sarcoplasmic Ca²⁺ and is functionally linked to ATP production, while calcium signaling at the nuclear membrane is regulated by distinct and still poorly understood mechanisms (Lines 483-484 and lines 569-576).